# Comparative Reverse Vaccinology of *Piscirickettsia salmonis*, *Aeromonas salmonicida*, *Yersinia ruckeri*, *Vibrio anguillarum* and *Moritella viscosa*, Frequent Pathogens of Atlantic Salmon and Lumpfish Aquaculture

**DOI:** 10.3390/vaccines10030473

**Published:** 2022-03-18

**Authors:** Joy Chukwu-Osazuwa, Trung Cao, Ignacio Vasquez, Hajarooba Gnanagobal, Ahmed Hossain, Vimbai Irene Machimbirike, Javier Santander

**Affiliations:** Marine Microbial Pathogenesis and Vaccinology Laboratory, Department of Ocean Sciences, Memorial University of Newfoundland, St. John’s, NL A1C 5S7, Canada; jchukwuosazu@mun.ca (J.C.-O.); ttcao@mun.ca (T.C.); ivasquezsoli@mun.ca (I.V.); hgnanagobal@mun.ca (H.G.); ahossain@mun.ca (A.H.); imachimbirik@mun.ca (V.I.M.)

**Keywords:** polyvalent, co-infection, antigen, outer membrane proteins (OMPs), secreted protein, structural homology

## Abstract

Marine finfish aquaculture is affected by diverse infectious diseases, and they commonly occur as co-infection. Some of the most frequent and prevalent Gram-negative bacterial pathogens of the finfish aquaculture include *Piscirickettsia salmonis*, *Aeromonas salmonicida*, *Yersinia ruckeri*, *Vibrio anguillarum* and *Moritella viscosa*. To prevent co-infections in aquaculture, polyvalent or universal vaccines would be ideal. Commercial polyvalent vaccines against some of these pathogens are based on whole inactivated microbes and their efficacy is controversial. Identification of common antigens can contribute to the development of effective universal or polyvalent vaccines. In this study, we identified common and unique antigens of *P. salmonis*, *A. salmonicida*, *Y. ruckeri*, *V. anguillarum* and *M. viscosa* based on a reverse vaccinology pipeline. We screened the proteome of several strains using complete available genomes and identified a total of 154 potential antigens, 74 of these identified antigens corresponded to secreted proteins, and 80 corresponded to exposed outer membrane proteins (OMPs). Further analysis revealed the outer membrane antigens TonB-dependent siderophore receptor, OMP assembly factor BamA, the LPS assembly protein LptD and secreted antigens flagellar hook assembly protein FlgD and flagellar basal body rod protein FlgG are present in all pathogens used in this study. Sequence and structural alignment of these antigens showed relatively low percentage sequence identity but good structural homology. Common domains harboring several B-cells and T-cell epitopes binding to major histocompatibility (MHC) class I and II were identified. Selected peptides were evaluated for docking with Atlantic salmon (*Salmo salar*) and Lumpfish MHC class II. Interaction of common peptide-MHC class II showed good in-silico binding affinities and dissociation constants between −10.3 to −6.5 kcal mol^−1^ and 5.10 × 10^−9^ to 9.4 × 10^−6^ M. This study provided the first list of antigens that can be used for the development of polyvalent or universal vaccines against these Gram-negative bacterial pathogens affecting finfish aquaculture.

## 1. Introduction

Bacterial infectious diseases are a common health concern and intractable economic issue affecting global finfish aquaculture [1]. Some of the most frequent marine bacterial pathogens identified in global aquaculture include *Aeromonas salmonicida* [2,3,4,5,6,7] *Moritella viscosa* [8,9], *Piscirickettsia salmonis* [10,11,12], *Vibrio anguillarum* [13,14] and *Yersinia ruckeri* [15,16,17,18,19,20]. Some of these pathogens have been reported to affect important fish species like Atlantic salmon and lumpfish [21,22,23,24,25,26].

Vaccination has been used successfully in finfish aquaculture for many decades to control diseases [27,28] and their use has increased significantly in recent years [29]. There are over 20 licensed vaccines for fish commercially available [30,31,32]. Most of these commercial vaccines are composed of whole inactivated pathogen and few are based on specific antigenic peptides, recombinant proteins, DNA vector vaccines, or live-attenuated microbes. The first vaccine for finfish was developed in 1942, based on whole inactivated microbes [33,34], and nowadays, the majority of licensed vaccines for teleost still use whole inactivated pathogens [30,31,32]. Some of these vaccines have successfully protected finfish against a variety of diseases like furunculosis, vibriosis, winter ulcer, piscirickettsiosis and enteric red mouth disease caused by *A. salmonicida*, *V. anguillarum*, *M. viscosa*, *P. salmonis* and *Y. ruckeri*, respectively [35]. However, several factors like the diversity of pathogen species and serotypes [36], co-infections [37], and diversity of cultured fish species pose a limitation to the efficacy of whole pathogen-based vaccines [36]. Specifically, whole inactivated microbe vaccines elicit a shorter length of protection and weak immune response because inactivated bacteria have been reported to induce modest cellular immunity to the fish [38,39]. Thus, there is a need for polyvalent vaccines that provide long-lasting immune protection against several pathogens to fish hosts. When vaccines only protect hosts from disease symptoms but allow for some level of pathogen infection and onward transmission, it can lead to the selection of more virulent strains [40,41,42]. Recombinant subunit vaccines, which are based on specifically targeted antigens, are more efficient in inducing humoral and cell-mediated immunological responses, and the risks associated with virulence evolution are reduced [43].

Multivalent vaccines used in Atlantic salmon (*Salmo salar*) aquaculture still employ the same conventional vaccine formulation based on whole-inactivate microbes [36]. For instance, Forte micro vaccine produced by Elanco contains formalin-inactivated cultures of *A. salmonicida*, *V. anguillarum* serotypes I and II, *V. ordalii* and *V. salmonicida* serotypes I and II in liquid emulsion with an oil-based adjuvant [44]. Alpha Ject micro 4 produced by Pharmaq contains formalin-inactivated cultures of *A. salmonicida* subsp. *salmonicida*, *Listonella* (*Vibrio*) *anguillarum* serotypes O1 and O2 and *V. salmonicida* [45]. The efficacy of these polyvalent vaccines based on whole inactivated microbes may be limited by the presence of immune-suppressive epitopes [36,40] and antigen competition or interference [46,47]. It is important to carefully select the number and properties of bacterial antigens in polyvalent vaccines to avoid inhibitory effects of antigens on the specific response of fish [48].

The development of an effective polyvalent or universal vaccine is dependent on the identification and selection of common protective antigens and epitopes [39,49,50]. The availability of complete bacterial genomes, including different strains and serotypes, and several in silico tools [49] make it possible to select common protective antigens that can be utilized in the development of effective universal or polyvalent vaccines for finfish [51]. Reverse vaccinology is a predictive bioinformatics analysis that can identify potential protective antigens and epitopes [51]. Reverse vaccinology was first used against *Neisseria meningitidis* serogroup B two decades ago after many unsuccessful trials using the conventional method of vaccine development [52,53]. Over 600 potential antigens were found in the sequenced genome of *N*. *meningitidis* serogroup B. It has been shown that predicted secreted or outer membrane proteins can serve as protective antigens [54,55,56,57,58]. Despite the success of the reverse vaccinology approach for vaccine development, this is yet to be fully exploited in the identification of novel antigens for fish vaccinology. Reverse vaccinology has been used to identify novel antigens against fish pathogens, including *V. anguillarum* [49] and *Photobacterium damselae* subsp. *piscicida* [59]. However, other major Gram-negative pathogens affecting the marine finfish aquaculture are yet to be fully explored.

Here, we developed a reverse vaccinology approach to screen the bacterial proteomes of the most frequent marine Gram-negative bacterial pathogens affecting the salmonid finfish aquaculture, including *P. salmonis*, *A. salmonicida*, *Y. ruckeri*, *V. anguillarum*, and *M. viscosa* completed genomes. We identified common antigens and epitopes that can be utilized in the development of universal or polyvalent vaccines. We evaluated the vaccine potentials of these common epitopes by visualizing their interaction with Atlantic salmon and Lumpfish major histocompatibility complex class II (MHC class II). These epitopes provide a library for the development of a polyvalent or universal vaccine against the marine pathogens considered in this study. The approach employed in this study can be used for the development of a universal vaccine against an even wider range of bacterial pathogens affecting the aquaculture industry.

## 2. Materials and Methods

### 2.1. Acquisition of Proteomes

Protein sequences from complete and annotated genomes utilized are listed in Appendix A (See Appendix A). *P. salmonis*, *A. salmonicida*, *M. viscosa*, *V. anguillarum* and *Y. ruckeri* proteomes were obtained in FASTA format from National Centre for Biotechnology Information (NBCI) public database (https://www.ncbi.nlm.nih.gov/) (accessed on 10 January 2022).

### 2.2. Subcellular Localization

Potential protective antigens must be able to interact with the host molecular receptor that will trigger a signal cascade and ultimately induce a protective adaptive immune response. Therefore, outer membrane proteins and secreted proteins were selected for further analyses based on the pipeline designed for this study (Figure 1). To identify outer membrane proteins and secreted proteins, the protein sequences retrieved from NCBI were analyzed for their subcellular localization using the dynamic Vaxign analysis (http://www.violinet.org/vaxign/) (accessed on 28 December 2021) [60,61,62] and PSORTb V.3.0.2 server (https://www.psort.org/psortb/) (accessed on 28 December 2021) [63]. The Vaxign web server utilizes the vaxign pipeline to dynamically calculate the possibilities of using the proteins as vaccine candidates [52]. Parameters were set to analyze subcellular localization, transmembrane helices, and adhesion probability. Self-antigen screening was performed on vaxign by matching the bacteria antigen to human antigens. Vaxign is not specifically trained to identify antigens for fish, therefore, fish MHC alleles are not included in the vaxign database. The best alternative for self-antigen screening was against human MHC alleles since they have the closest sequence identity to fish MHCs [49]. The subcellular locations assessed were outer membrane, cytoplasmic membrane, cell wall, periplasmic and extracellular regions. The subcellular localization probability scores range from 0 to 1 on Vaxign and 0 to 10 on PSORTb for each cellular location. Only proteins localized on the outer membrane and extracellular regions that have transmembrane helices score of ≤ 1 and no similarity to eukaryotes (human) MHC were selected for further analysis.

### 2.3. Adhesin and Antigenic Probability Scores

Adhesin probability scores were generated using Vaxign [60,61,62] (http://www.violinet.org/vaxign/) (accessed on 10 January 2022) with a threshold of ≥0.51, while the antigenic probability scores were calculated using VaxiJen V.2.0 (http://www.ddg-pharmfac.net/vaxijen/VaxiJen/VaxiJen.html) (accessed on 10 January 2022) with a threshold of ≥0.4 [64,65]. The protein sequences of selected putative vaccine candidates were saved in FASTA format for further analysis.

### 2.4. Sequence and Structural Alignment

The multiple sequence alignment for the common antigens of the bacterial pathogens was conducted using Clustal 2.1 (clustal omega) with default parameters set to generate a percent identity matrix [66]. The tridimensional protein structure of selected antigens was modeled using HHpred [67]. Structural Alignment of Multiple Proteins (STAMP) on Visual molecular dynamics (VMD) was further used to compare the tridimensional protein structures of the antigens (http://www.ks.uiuc.edu/Research/vmd/) (accessed on 10 January 2022). STAMP aligns protein sequences based on a three-dimensional structure. Its algorithm minimizes the alpha carbon (Cα) distance between aligned residues of each molecule by applying globally optimal rigid-body rotations and translations [68,69]. This structural alignment method requires that multiple alignments should be used for a single domain evolutionary analysis and not for the alignment of larger multi-domain proteins unless the multi-domain proteins display homology over all domains. Any attempt to align unrelated multiple domain structures or proteins that have no similarities with STAMP may result in alignment failure [68]. The structural homology (QH) values, root mean square deviation (RMSD) and percent identity were generated using Visual Molecular Dynamic (VMD) tools. Structural conservation between proteins (Q) was determined by calculating Structural homology between protein structures [70]. Structures that showed a Q = 1 were considered identical. Structural alignment with Q values <0.3 was considered not well-aligned [70].

### 2.5. Identification of Putative Domains

To further characterize the selected antigens, the presence of putative domains for all common antigens identified in this study was generated by querying the sequence against NCBI conserved domain database (CDD) using the position-specific iterated (PSI) BLAST.

### 2.6. Detection of B and T Cell Epitopes

B-cell epitopes were predicted for selected antigens using tools from the Immune Epitope Database (IEDB) analysis resource BepiPred linear epitope prediction [71]. The BepiPred linear epitope predicts the location of linear B-cell epitopes using a combination of a hidden Markov model and a propensity scale method [71]. The residues with scores > 0.35, sensitivity > 0.49 and specificity > 0.75 were considered to be part of a B cell epitope [71]. Vaxitop, a tool in the Vaxign server [60,61] was used to identify T cell epitopes binding to human MHCs (Appendix A). A *p*-value of ~0.05 was used as a cutoff.

### 2.7. Atlantic Salmon and Lumpfish Major Histocompatibility Homology Modelling

Homology modeling of Atlantic salmon and Lumpfish Major Histocompatibility complex class II (MHC class II) was carried out on SWISS-MODEL webserver (https://swissmodel.expasy.org/) (accessed on 10 January 2022) [72]. Atlantic salmon MHC II α (Q5ZQM4_SALSA) and β (Q95IS0_SALSA) chains were retrieved from the SWISS-MODEL repository. FASTA files of the protein sequence of Lumpfish MHC II α (Accession number: XP_034409868.1) and β (Accession number: XP_034409871.1) chains were uploaded to the webserver program and the tridimensional models were generated. Subsequently, the obtained MHC class II chains were docked together to produce the MHC II complex using the ClusPro V2.0 webserver (https://cluspro.bu.edu/login.php) (accessed on 10 January 2022) [73]. All parameters were set to default settings for the protein-protein docking. The quality of the model was checked using the Procheck webserver (https://www.ebi.ac.uk/thornton-srv/software/PROCHECK/) (accessed on 10 January 2022) [74].

### 2.8. Epitope Docking of Common Epitopes with Atlantic salmon and Lumpfish MHC Complex

To date, all the available programs to predict epitopes based on MHC complexes do not contain fish MHC alleles. Therefore, in this study, human MHC alleles were used to predict the epitopes. However, it has been reported that human MHC alleles have low sequence identities with fish MHC alleles of ~35 to 38% to human MHC sequences [49]. To validate that the predicted common epitopes identified in this study can bind to the Atlantic salmon MHC class II, protein–peptide docking was performed using the online program CABS-dock (http://biocomp.chem.uw.edu.pl/CABSdock) (accessed on 10 January 2022). This program performs protein-peptide docking using simulation to search for the binding site on the receptors, which allows for full flexibility of the peptide as well as small fluctuations of the receptor backbone [75]. The online webserver PRODIGY (https://wenmr.science.uu.nl/prodigy/) (accessed on 10 January 2022) was then used to predict binding affinities and dissociation constants of the protein-peptide complexes [76].

## 3. Results

### 3.1. Pipeline Development

The reverse vaccinology pipeline used in this study was categorized into selection and characterization phases, respectively (Figure 1). The selection phase consisted of bacterial proteome acquisition, screening, and selection of common antigens. A total of 5 in-silico tools and one manual filtration were used to achieve this. The characterization phase involved a further description of the common antigens, epitopes and investigating the degree of their similarity across the pathogens used in this study (Table 1). A total of 9 in-silico tools were employed for the characterization phase (Figure 1).

### 3.2. Subcellular Localization

The Vaxign subcellular localization analysis (Figure 2) revealed the number of proteins localized in various bacterial cellular regions. As expected, the cytoplasmic region has the highest number of proteins, while outer membrane proteins (OMPs) and secreted proteins were less abundant. On average, we detected approximately 3146 cytoplasmic proteins, 76 OMPs, and 57 secreted proteins for all bacterial proteomes analyzed in this study. *M. viscosa* showed the highest number of OMPs and secreted proteins, in contrast to *P. salmonis* which showed the lowest OMPs and *V. anguillarum*, which showed the lowest number of secreted proteins.

### 3.3. Adhesin and Antigenicity

The selected OMPs and secreted proteins were evaluated for their ability to function as adhesins and for their antigenicity index, which makes them potential vaccine candidates [49]. The Vaxign vaccine design and Vaxijen informatic packages generated adhesin and antigenicity scores, respectively, for the OMPs and secreted protein. A total 161 immunogenic proteins were identified in this study, 80 of them are secreted proteins, and 81 are exposed OMPs. Hypothetical proteins, which are proteins not yet characterized on NCBI, were not included in this analysis. Based on the description of the immunogenic proteins identified, some proteins are common to the bacterial pathogens used in this study, while others are unique to each bacteria pathogen (Figure 3). The outer membrane protein assembly factor (BamA), the LPS assembly protein (LptD), and dependent siderophore receptor (TonB) are the 3 common immunogenic proteins identified from the outer membrane proteins across all bacteria proteomes, while flagellar hook assembly protein FlgD and flagellar basal-body rod protein were the two common antigens identified from the secreted immunogenic proteins identified (Figure 3 and Table 2). Outer membrane transport protein was a common antigen identified in *P. salmonis*, *A. salmonicida*, *V. anguillarum* and *M. viscosa* but absent in *Y. ruckeri*. Porin, hemoglobin/transferrin/lactoferrin family receptor and flagellar basal body L-ring protein (FlgH) were common antigens identified in *A. salmonicida*, *V. anguillarum*, *Y. ruckeri* and *M. viscosa* but absent in *P. salmonis*. Flagellar filament capping protein FliD was a common antigen identified in *P. salmonis*, *A. salmonicida*, *Y. ruckeri* and *M. viscosa* but absent in *V. anguillarum.* Other proteins include the OmpA family protein, outer membrane protein assembly factor BamE, TolC family outer membrane protein, domain-containing proteins, TraV, maltoporin, TonB-dependent receptor, DUF2860 family protein, membrane protein, M23 family metallopeptidase, carbohydrate porin, conjugal transfer protein TraF, murein hydrolase activator NlpD, outer membrane beta-barrel protein, MipA/OmpV family protein, flagellar hook-associated protein FlgK and FlgL, flagellin, phage tail, and others were common to two or three bacteria pathogens (Appendix A).

### 3.4. Comparative Topology

Multiple sequence alignments of the common antigens identified in this study showed low percentage identity but high structural homology across the various bacterial pathogens (Table 3 and Table 4). The 3-dimensional secondary structures of the identified common antigens (OMP assembly factor BamA, TonB dependent siderophore receptor, LPS assembly protein LptD, flagellar basal-body rod protein FlgG and flagellar hook assembly protein (FlgD) showed similar structures (Figure 4 and Figure 5). The structural homology (Q_H_) values of the common antigens range from 0.24 to 0.88 with corresponding RMSD values ranging from 0.90 to 6.27, indicating a good structural homology (Table 3 and Table 4). Furthermore, the PSI-BLAST analysis of the protein sequence for these common antigens revealed that they have the same conserved domains across the bacterial pathogens (Appendix A).

### 3.5. Epitope Prediction and Docking to Fish MHC Complexes

A total of 64 epitopes was predicted from the common proteins identified using *P. salmonis* LF 89 as the model for this epitope prediction (Appendix A and Figure 6). Sequence alignment revealed 13 epitopes are conserved across the bacterial pathogens studied (Figure 7 and Table 5). These common epitopes were selected for protein-peptide docking with Atlantic salmon and Lumpfish MHC II. The model for alpha and beta chains for Atlantic salmon MHC II (Figure 8) showed 91.3% (336/368) of all residues in favored (98%) regions and 98.9% (364/368) of all residues in allowed (>99.8%) regions. Protein–peptide docking of the common epitopes with Atlantic salmon MHC II (Figure 8) showed acceptable predicted binding affinities ranging from −10.3 to −6.5 kcal mol^−1^ and dissociation constants ranging from 5.10 × 10^−9^ to 9.4 × 10^−6^ M (Table 5). Protein–peptide docking of the common epitopes with lumpfish MHC II (Appendix A) showed acceptable predicted binding affinities ranging from −10.7 to −6.5 kcal mol^−1^ and dissociation constants ranging from 5.0 × 10^−9^ to 9.4 × 10^−6^ M (Table 5).

## 4. Discussion

Bacterial pathogens affecting finfish aquaculture are relentless, and their impact is enhanced by co-infections [37] and the increasing diversity of pathogen species and serotypes [36]. Commercial polyvalent vaccines available in the aquaculture industry are not providing optimal protection in all finfish species. This comparative reverse vaccinology study aimed to identify common antigens and epitopes across various marine Gram-negative bacteria affecting finfish aquaculture and provide crucial information for the development of effective polyvalent or universal vaccines to prevent bacterial infectious diseases outbreaks in aquaculture.

Gram-negative pathogens synthesize several extracellular and secreted proteins including virulence factors like fimbria, secretion systems, and toxins essential for pathogenicity that can serve as effective antigens [77,78]. In this study, we identified antigens that are exposed onto the bacteria outer membrane or secreted to the extracellular milieu (Appendix A). Coincidently, some of these proteins such as BamA, LptD, TonB, FlgD, FlgG, aerolysin, hemolysin have been shown to confer immune protection in fish (e.g., Olive flounder, Zebrafish, Catfish, Grouper fish, Turbot,) against *Edwardsiella tarda*, *Y. ruckeri*, *V. anguillarum*, *V. alginolyticus* and *A. salmonicida* [58,79,80,81]. Adhesin and antigenic characteristics are also important features of a potential vaccine candidate [49]. Secreted and outer membrane proteins that scored adhesin and antigenic probability values of ≥ 0.51 and ≥ 0.4, respectively, were considered for the development of potential vaccine candidates in this study (Table 2). Five common antigens, including the LPS assembly protein LptD, the outer membrane protein assembly factor BamA, the TonB-dependent siderophore receptor, the flagellar hook assembly protein FlgD, and the flagellar basal-body rod protein FlgG were identified to be conserved across all the strains and species used in this study (Figure 1). These antigens are identical proteins on NCBI across strains of the same species, but a low percentage identity of the antigens across strains of the same species but a low percentage identity of the antigens across different bacterial species used in this study (Table 3 and Table 4). However, we hypothesized that these antigens might have important structural or functional similarities because they have the same protein description (Table 2). Since structural alignments are usually more specific to protein and are more evolutionarily conserved than the amino acid sequence [82], further analyses were conducted to investigate the extent of their structural homology using STAMP on VMD. STAMP has been used to identify the structurally conserved epitope “HAFYLQYKNVKVDFA” among superoxide dismutase homolog structures of *Mycobacterium tuberculosis* (superoxide dismutase 1IDS), *Aspergillus fumigatus* (Mn superoxide dismutase 1KKC), and human (Mn superoxide dismutase 2QKC) associated with autoimmune atopic dermatitis [83]. The common proteins identified across all bacterial pathogens in this study were successfully aligned by STAMP on VMD (Figure 4 and Figure 5), which is an indication of homology in their domains. This finding is further supported by the results of the (PSI)-BLAST of the common proteins identified in this study, which showed the presence of similar domains (Appendix A). Further analysis of the common antigens revealed a total of 64 novel B cells and T cells epitopes binding to several MHC class I and II alleles (Appendix A). Sequence alignment of these antigens across *P.*
*salmonis A. salmonicida*, *M. viscosa*, *Y. ruckeri* and *V. anguillarum* revealed external domains harboring common epitopes (Figure 7 and Table 5) across these bacterial pathogens. These epitopes were predicted using human MHC alleles because the database does not contain fish MHC alleles for teleost. Thus, to evaluate if these epitopes can bind to Atlantic salmon MHC class II, protein-peptide docking was performed to calculate the binding affinities and dissociation constants (Table 5). This analysis showed that these epitopes can successfully bind to MHC class II from Atlantic salmon and could be used for the polyvalent vaccine design against furunculosis, vibriosis, winter ulcerative disease, piscirickettsiosis, and enteric red mouth disease.

The common antigens in this study play important roles in bacteria cellular activities and are highly conserved in Gram-negative bacteria. Outer membrane protein assembly factor BamA is a key component of the beta-barrel assembly machinery (BAM) complex, which is a key player in OMP assembly [84]. PSI-blast of BamA revealed 5 flexible polypeptide-transport-associated (PORTA) domains (Appendix A), which guide substrates into the barrel scaffold and facilitate β-strand formation and assembly. BamA has been identified in a previous study as a potential vaccine candidate against *Acinetobacter baumannii* [85]. Mice intraperitoneally immunized with 20 µg BamA formulated with 2% Al (OH)_3_ adjuvant and intranasally challenged with 10^9^ CFU dose^−1^ of *A. baumannii* 45 days post-immunization, *showed* 80% survival. In addition, serum from BamA immunized animals used in passive immunization assays conferred 60% protection to mice after intranasal challenge with a lethal dose of *A. baumannii*. *A. baumannii* was not detected in lungs of BamA immunized mice after 7 days post-challenge. Reduction of pro-inflammatory cytokines (TNF-α, IL-6 and IL-1β) and increase in levels of anti-inflammatory cytokine (IL-10) were also observed [85]. These results suggest that BamA might be a good vaccine candidate and could protect the fish against the bath challenge.

LptD has been characterized in bacteria as an essential and well conserved outer membrane protein that mediates the final transport of lipopolysaccharide (LPS) to outer leaflet [86]. Outer membrane protein assembly factor BamA and LPS assembly protein LptD has been identified in silico as a promising vaccine candidate against *Moraxella catarrhalis* responsible for respiratory tract infections and middle ear infections in children and adults [87]. Numerous potential gonorrhea vaccine targets including BamA and LptD have been identified using proteomics-driven discovery [88]. LptD has also been identified as an immunogenic protein in *Vibrio parahaemolyticus* and *Salmonella enteritca* Serovar Typhi using immunoproteomics [89,90]. Subsequently, LptD has been shown to confer 100% protection against a lethal intraperitoneal challenge with 10^7^ CFU dose^−1^ of *V. parahaemolyticus* in mice and clearance of the bacteria in vaccinated mice [86].

TonB-dependent siderophore receptor mediates the transport of siderophores into the periplasm in Gram-negative bacteria [91]. Other iron-related antigens identified though not common across all strains studied here, include the siderophore amonabactin TonB-dependent receptor and TonB-dependent hemoglobin/transferrin/lactoferrin. TonB-dependent siderophore receptor has been described as an immunogenic protein because of the role it plays in iron transport during pathogenesis which is essential in the virulence of bacteria [92,93]. The potential of the TonB receptor as a vaccine candidate has been explored against some Gram-negative bacteria pathogens. For instance, flounder immunized with 100 μL purified recombinant TonB receptor showed strong protective immunity with a relative percent of survival (RPS) of 80.6% and production of serum-specific antibodies against a lethal dose of *P. fluorescens* challenge [92]. The protective properties of *Burkholderia mallei* attenuated strain which is a TonB mutant deficient in iron acquisition, have been evaluated in acute inhalational infection models of murine glanders and melioidosis, and it demonstrates great potential as a backbone strain for future vaccine development against both infections [94]. HgbA, a TonB-dependent hemoglobin receptor in *Haemophilus ducreyi* has been implicated in the virulence of this pathogen in humans and has the potential to be used in vaccine development [95]. In another study, HgbA showed immune protection against chancroid infection in swine model challenged, with 5 × 10^4^ CFU of *H. ducreyi* 3 weeks post-vaccination [96]. These studies support our findings and indicate that TonB-dependent siderophore receptors are a potential candidate for fish vaccine development.

Common secreted antigens identified are the Flagellar basal-body rod protein FlgG and flagellar hook assembly protein FlgD. Flagellins have been shown to bind to the Toll-like receptor 5 (TLR5), which induces innate immune system, triggering a cascade of signaling pathways response [97]. FlgG and FlgD have been explored as potential vaccine candidates in various experiments. Flagellar basal-body rod protein FlgG used in a recombinant vaccine with chaperones Hsp60 and Hsp70 showed high protection, with a relative percent survival (RPS) of 95% against piscirickettsiosis in Atlantic salmon [98].

Olive flounder (*Paralichthys olivaceus*) challenged with 5 × 10^6^ CFU dose^−1^ of *Edwardsiella tarda* after vaccination with oral vaccines expressing OmpA-FlgD, FlgD, and OmpA were protected from edwardsiellosis, with survival rates of 82.5%, 55% and 50%, respectively. In this experiment, flounder fed the FlgD-expressing vaccine, showed a significantly increased expression of TLR5M, IL-1β, and IL-12p40, suggesting that the FlgD may be a ligand of olive flounder TLR5M receptor or closely related to the TLR5M pathway [99]. Zebrafish and turbot showed high cellular mediated immune response with high expressions of MHC class I, MHC class II, CD4 and CD8ά genes, which are indications of T cells activation to flagellar hook assembly protein FlgD against *E. tarda* with RPS of 70% [100]. These studies support our findings and indicate that secreted antigens Flagellar basal-body rod protein FlgG and flagellar hook assembly protein FlgD are potential vaccine candidates that should be explored in fish vaccinology.

An ideal vaccine should induce both innate immunity and long-term stimulation of humoral and cell-mediated adaptive immunity by producing effector and memory cells [101]. This immune response could be achieved by using antigens that possess high epitope density [102]. In this study, 13 common epitopes were identified in the common antigens across the five bacteria species analyzed that could be used as part of polyvalent vaccines (Table 5 and Figure 9). Most in-silico tools for reverse vaccinology currently available are not adapted for fish vaccinology. For example, Vaxitop software used to predict T cell epitopes is not specifically trained to predict peptides based on binding to the fish MHC alleles, but based on binding to human and other mammalian MHC alleles, like mice [49]. In the absence of fish MHC alleles to predict epitopes, epitope predictions were conducted using human alleles, which have an identity of ~35 to 38% to fish MHC sequences [49]. Atlantic salmon and lumpfish MHC class II was modeled (Figure 8 and Appendix A), and a protein-peptide docking was performed with the identified common epitopes to confirm binding to MHC class II (Figure 9). Binding affinity was considered high when Kd < 0.1 nM (1.0 × 10^−10^), medium when Kd is between 0.1 nM to 1 μM (1.0 × 10^−10^–1.0× 10^−6^), and low when Kd >1 μM [103]. Furthermore, according to thermodynamics, the binding of a ligand/epitope to a protein only occurs when the system’s change in the Gibbs free energy (ΔG) is negative and the magnitude indicates the stability and binding affinity of the protein-ligand complex [104]. Our results showed that all the predicted epitopes could bind to lumpfish MHC class II complex with high negative ΔG values with medium to high binding affinities. Hence, the predicted epitopes have a high potential of being recognized by Atlantic salmon and Lumpfish MHC class II to elicit a humoral response in lumpfish.

Aside from the common antigens identified, several other antigens were common to two, three or four bacteria species used in this study. Some of these antigens have been reported in other studies to be potential vaccine candidates for fish or other animals. For instance, OmpA was identified to be common in *P. salmonis*, *M. viscosa*, *A. salmonicida* (as porin OmpAI and OmpAII), and *V. anguillarum* (as porin OmpA). OmpA has been reported to provide 73.3% survival to *Labeo fimbriatus* that were orally vaccinated with recombinant OmpA encapsulated in chitosan nanoparticles and challenged against *E. tarda* [105]. A new hybrid OmpA developed by DNA shuffling was demonstrated as an ideal polyvalent vaccine against infections caused *V. alginolyticus* and *E. tarda* [106]. Olive flounder fed feeds expressing OmpA, flagellar FlgD, or a fusion antigen of the two for one week showed 55%, 50% and 82.5%, respectively [99]. OmpA protein has been characterized and observed to be highly immunogenic in fish. Common carp vaccinated with recombinant OmpA protein showed a relative percentage survival of 54.3% against *E. tarda* challenge and elicited a high antibody production in immunized fish [107]. Another antigen identified in this study to be common in up to 3 bacteria species (*P. salmonis*, *V. anguillarum* and *M. viscosa*) and has been reported in other studies to be potential vaccines candidate is the TolC family outer membrane protein. TolC has been identified to be good vaccine candidates against *E. tarda* and *F. columnare* in a computer aided vaccine design [108]. Purified recombinant TolC has been shown to confer significant protection with a RPS of 73.68% in hybrid grouper fish against *V. harveyi* challenge and competitively inhibit the invasion of *V. harveyi* to grouper embryonic cells in vitro [109]. The M23 family metallopeptidase identified in *V. anguillarum* and *A. salmonicida* was evaluated as potential candidate for a broad-spectrum vaccine against *Neisseria meningitidis*, the causative agent of meningococcal disease in humans [110]. Murein hydrolase activator NlpD identified in *Y. ruckeri* and *V. anguillarum* has been reported to be essential for *Y. pestis* virulence and the *nlpD* mutant can serve as a basis for developing a potent live vaccine against bubonic and pneumonic plague [111]. OmpV (MipA) family protein identified in *Y. rucker* and *V. anguillarum* has been reported to induce high IgG production and protection against *Salmonella enterica* serovar Typhimurium systemic and gastrointestinal disease in mice. Additionally, it has been shown that OmpV activates innate immune cells, such as monocytes, macrophages, and intestinal epithelial cells, in a Toll-like receptor 2-dependent manner [112]. OmpV can be recognized by both TLR1/2 and TLR2/6 heterodimers, this increases its potential to act as a good adjuvant in other vaccine formulations [112].

Several bacterial toxins and virulent factors identified in this study are unique to specific bacteria species and can be explored in the development of a monovalent or polyvalent multiepitope vaccine in fish vaccinology. Some of these antigens have been reported to be potential vaccine candidates. For instance, hemolysin and aerolysin are extracellular virulent factors identified in *V. anguillarum* and *M. viscosa*, respectively. Catfish immunized with recombinant hemolysin (rHly) and aerolysin induced humoral immune response as evidenced by immunoblotting and cell agglutination; immunized catfish showed 71–78% survival when challenged with virulent *A. hydrophila* and monitored for 2–5 weeks [113]. In another experiment, catfish immunized with the T6SS effector, hemolysin co-regulated protein (Hcp) resulted in an increased survival (46.67%) in common carp during a 10-day challenge time compared to non-vaccinated fish (7.14%). The vaccinated fish also showed significantly increased levels of IgM antibody in serum and cytokines (interleukin-1β and tumor necrosis factor-α) in the kidney, spleen and gills [114]. Mass spectrometric analysis of extracellular products (ECP) from *A. hydrophila* that conferred 100% protection to catfish vaccinated with ECP when challenged with the pathogen two weeks post-vaccination showed several putative proteins that may serve as important immunogens, including chitinase, chitodextrinase, outer membrane protein 85, putative metalloprotease, extracellular lipase, hemolysin, and elastase [115]. Other enzymes like collagenase, lytic polysaccharide monooxygenase, sphingomyelin phosphodiesterase, methyltransferase, triacylglycerol lipase, exo-alpha-sialidase, deoxyribonuclease I, nucleotidyltransferase, class C beta-lactamase and SGNH/GDSL hydrolase family protein identified in this study as potential vaccine candidates have been implicated in the virulence of bacterial pathogen [116,117,118,119,120,121,122,123,124,125], and could be considered for a design of an effective polyvalent vaccine for finfish.

## 5. Conclusions

There is a need for effective polyvalent vaccines in aquaculture that prevent the emergence of novel virulent variants and reduce co-infections. The increasing number of bacterial genomes available and tools for reverse vaccinology offers a platform for the identification of novel common and uncommon antigens across bacterial pathogens that allows the development of a potential universal anti-bacterial vaccine. We have provided a reverse vaccinology pipeline to identify common and unique antigens across strains of frequent marine bacterial pathogens of finfish, including *P. salmonis*, *A. salmonicida*, *Y. ruckeri*, *V. anguillarum* and *M. viscosa.* Potential B cell and T cell epitopes from these common antigens were also identified and docked to Atlantic salmon and Lumpfish MHC class II. Based on these analyses, we identified five common antigens, which contain 13 exposed epitopes that could interact with CD4^+^ T and B cells. These epitopes could be considered as a base to develop an anti-bacterial polyvalent vaccine for finfish, but additional epitopes from individual antigens need to be incorporated to complete a possible fully functional universal vaccine. In addition, docking with other fish species would be needed to ensure broad immunogenicity. This study contributes to the antigen selection and design of a universal vaccine across a wider range of bacteria affecting the aquaculture industry.

## Figures and Tables

**Figure 1 vaccines-10-00473-f001:**
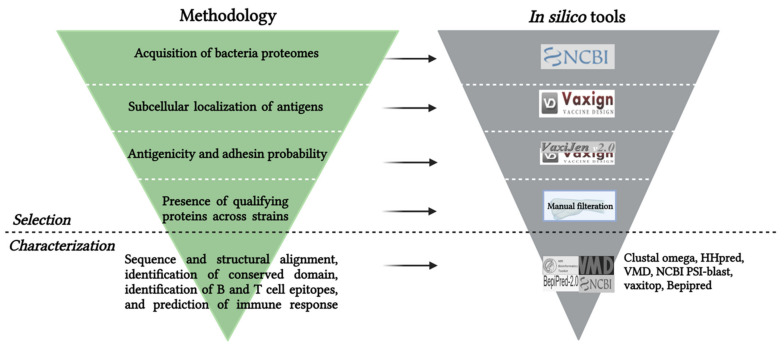
Pipeline used for this study to identify immunogenic proteins and epitopes of marine bacterial pathogens.

**Figure 2 vaccines-10-00473-f002:**
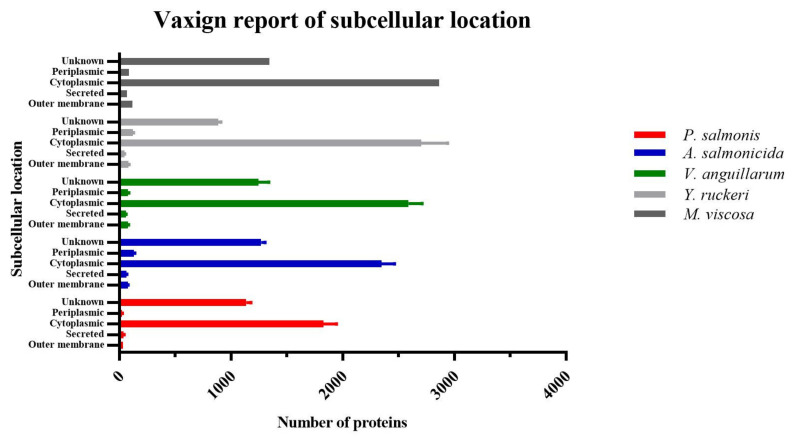
Vaxign report on subcellular localization of antigens discovered in all proteomes analyzed (Table 1). *Moritella viscosa* has no standard deviation (SD)bar because only one proteome was available for this study.

**Figure 3 vaccines-10-00473-f003:**
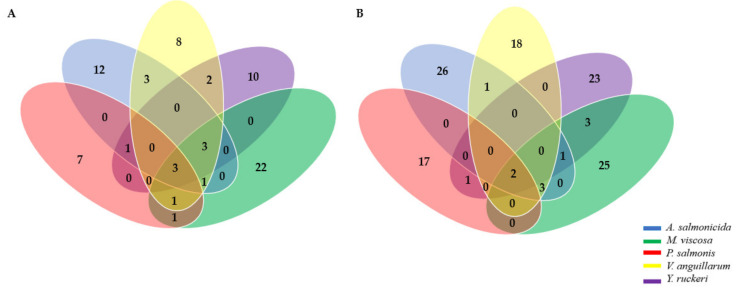
Immunogenic proteins identified across all bacterial proteomes from marine pathogens of fish used in this study. (**A**) Outer membrane proteins; (**B**) Secreted proteins.

**Figure 4 vaccines-10-00473-f004:**
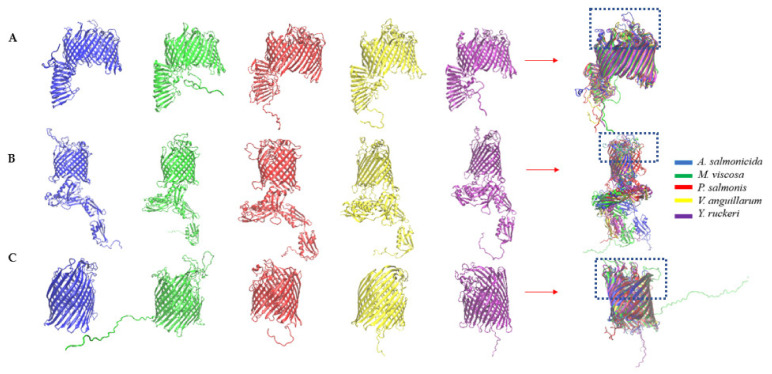
Three-dimensional secondary structures of common Outer membrane antigens showing structural homology for bacterial pathogens used in this study. (**A**) LPS-assembly protein LptD; (**B**) outer membrane protein assembly factor BamA protein; (**C**) TonB-dependent siderophore receptor. The regions of conserved domains are highlighted in the boxes (see Appendix A for a complete list of the conserved regions highlighted).

**Figure 5 vaccines-10-00473-f005:**
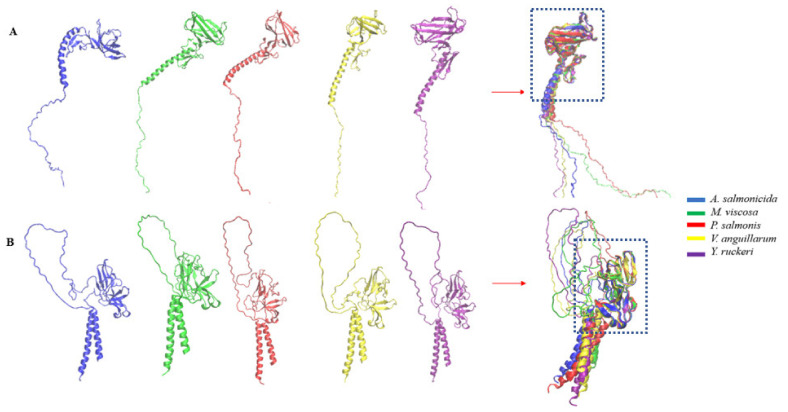
3D secondary structures of common secreted antigens showing structural homology for bacterial pathogens used in this study. (**A**) flagellar hook assembly protein FlgD; (**B**) flagellar basal-body rod protein FlgG. The regions of conserved domains are highlighted in the boxes (see Appendix A for a complete list of the conserved regions highlighted).

**Figure 6 vaccines-10-00473-f006:**
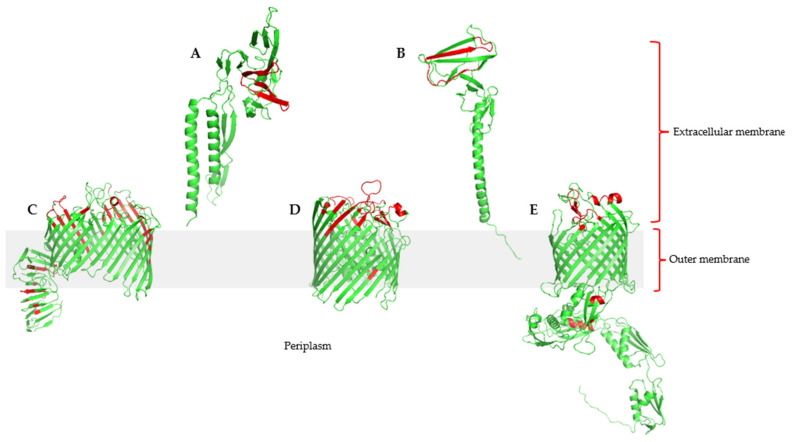
3D visualization of the common outer membrane and secreted antigens showing predicted B and T cell epitopes. (**A**) Flagellar hook assembly protein FlgD; (**B**) flagellar basal-body rod protein FlgG; (**C**) LPS-assembly protein LptD; (**D**) outer membrane protein assembly factor BamA protein; (**E**) TonB-dependent siderophore receptor. Immune epitopes are highlighted in red.

**Figure 7 vaccines-10-00473-f007:**
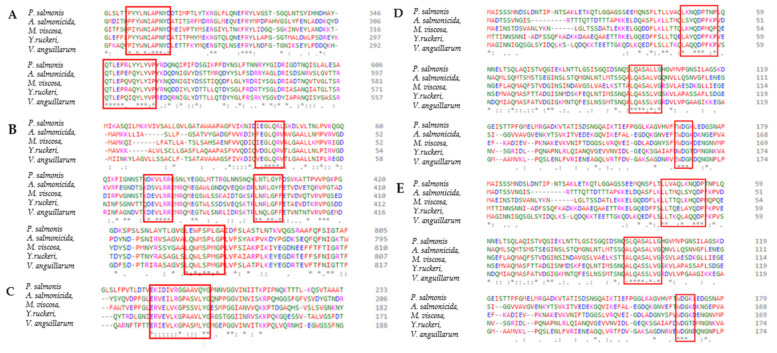
Sequence alignment showing domains harboring common epitopes across the bacterial pathogens studied. (**A**) LPS assembly protein LptD; (**B**) outer membrane protein assembly factor BamA; (**C**) TonB-dependent siderophore receptor; (**D**) flagellar hook assembly protein FlgD; (**E**) flagellar basal-body rod protein FlgG.

**Figure 8 vaccines-10-00473-f008:**
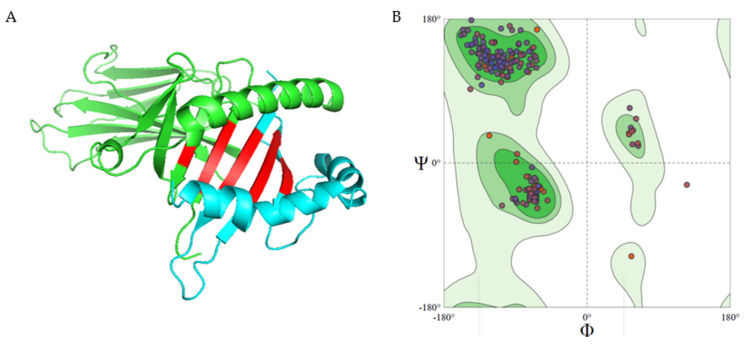
Modeling of Atlantic salmon MHC class II complex. (**A**) 3D structure of Atlantic salmon MHC class II complex, this comprises of MHC class II α chain (green), MHC class II β chain (cyan) and the epitope binding pocket (red); (**B**) Ramachandran plot showing the quality of salmon MHC class II complex modelled, 98.4% (251/255) of all residues were in favored (98%) regions. 98.8% (252/255) of all residues were in allowed (>99.8%) regions. There were three outliers (phi, psi): A 73 Tyr (125.4, −27.3) A 131 Pro (62.1, 120.2) A 160 Val (63.8, 24.5) (also see Appendix A).

**Figure 9 vaccines-10-00473-f009:**
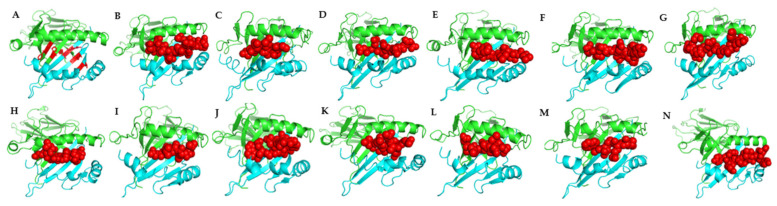
Interactions between Atlantic salmon MHC class II complex and the common epitopes. (**A**) MHC class II complex of Atlantic salmon; (**B**) PYYLNLAPNYD epitope docked to MHC class II; (**C**) QTLEPRLYYLYVP epitope docked to MHC class II; (**D**) IEGLQRL epitope docked to MHC class II; (**E**) DEVLARE epitope docked to MHC class II; (**F**) LNTLGYF epitope docked to MHC class II; (**G**) LQWMSPLGP epitope docked to MHC class II; (**H**) EKIDVRGGAAVQYG epitope docked to MHC class II; (**I**) LKNQDPPTNP epitope docked to MHC class II; (**J**) LQASALLG epitope docked to MHC class II; (**K**) WDGNDQNGN epitope docked to MHC class II; (**L**) LLTQLAQQDP epitope docked to MHC class II; (**M**) ALQASALVG epitope docked to MHC class II; (**N**) WDGKL epitope docked to MHC class II; The green domain represents the MHC class II α chain, the cyan domain represents the MHC class II β chain and red peptides for the epitopes. Hydrogen bonds were identified between the epitopes and MHC class II molecules.

**Table 1 vaccines-10-00473-t001:** Bacterial strains used for this study.

Species	Genomes Used in This Study (NCBI Accession Numbers)
*Piscirickettsia salmonis*	LF 89, EM 90, AY3800B, AY3864B, AY6297B, PM25344B, AY6532B, PM31429B, PM32597B1, PSCGR01, PM49811B, PM58386B, PM22180B, AY6492A, PM15972A1, PM23019A, PM37984A, PM51819A, PM21567A.
*Aeromonas salmonicida*	J223, A449, O23A, S121, S68, S44, RFAS1, 34mel, A527, 01-B526.
*Moritella viscosa*	MVIS1
*Yersinia ruckeri*	SC09, Big Creek 74, YRB, QMA0440, NHV_3758
*Vibrio anguillarum*	JLL237, NB10, MVAV6203, ATCC-68554, M3, 90-11-286, S3 4/9, MHK3, VIB43

**Table 2 vaccines-10-00473-t002:** Common potential vaccine candidates identified across all bacteria pathogens.

Bacteria	Accession Number	Common Antigenic Proteins	Antigenic Probability Cutoff (0.4)	Adhesin Probability Cutoff (0.51)	Location
*P. salmonis* *A. salmonicida* *M. viscosa* *Y. ruckeri* *V. anguillarum*	WP_036771893 WP_011898993 WP_045111816 WP_038245171 WP_019282310	Outer membrane protein assembly factor BamA	0.484 0.656 0.533 0.577 0.559	0.811 0.613 0.608 0.716 0.612	Outer membrane
*P. salmonis* *A. salmonicida* *M. viscosa* *Y. ruckeri* *V. anguillarum*	WP_005316218 WP_017377460 CED58662 WP_080748387 WP_019282793	TonB dependent siderophore receptor	0.567 0.606 0.599 0.576 0.631	0.832 0.497 0.525 0.424 0.714	Outer membrane
*P. salmonis* *A. salmonicida* *M. viscosa* *Y. ruckeri* *V. anguillarum*	WP_048876074 WP_005311708 CED61712 WP_004721574 WP_029388238	LPS assembly protein LptD	0.514 0.727 0.600 0.546 0.580	0.739 0.517 0.472 0.671 0.740	Outer membrane
*P. salmonis* *A. salmonicida* *M. viscosa* *Y. ruckeri* *V. anguillarum*	WP_027242990 WP_005320202 CED58793 WP_004720726 WP_010319379	Flagellar basal-body rod protein FlgG	0.593 0.628 0.477 0.579 0.578	0.764 0.725 0.751 0.791 0.723	Secreted
*P. salmonis* *A. salmonicida* *M. viscosa* *Y. ruckeri* *V. anguillarum*	WP_027242992 WP_005320209 CED61365 WP_038242029 WP_013856252	Flagellar hook assembly protein FlgD	0.405 0.633 0.547 0.555 0.466	0.510 0.795 0.780 0.710 0.606	Secreted

**Table 3 vaccines-10-00473-t003:** Sequence and structural alignment of the common outer membrane antigens. QH, structural homology; RMSD, root mean square deviation.

	Outer Membrane Protein Assembly BamA			LPS-Assembly Protein LptD			TonB-Dependent Siderophore Receptor		
Bacteria Pathogens	QH	RMSD	Identity (%)	QH	RMSD	Identity (%)	QH	RMSD	Identity (%)
*M.viscosa* vs *A. salmonicida*	0.52	4.42	41.80	0.71	2.35	30.69	0.64	1.96	20.20
*M. viscosa* vs *P. salmonis*	0.49	4.62	22.62	0.63	2.72	21.06	0.55	2.51	11.85
*M.viscosa* vs *V. anguillarum*	0.26	5.68	14.05	0.76	1.89	29.31	0.63	2.03	18.10
*M.viscosa* vs *Y.ruckeri*	0.25	6.27	14.75	0.74	1.80	29.84	0.84	0.82	19.50
*A.salmonicida* vs *P. salmonis*	0.67	2.06	27.41	0.72	1.63	21.74	0.58	2.69	14.05
*A.salmonicida* vs *V. anguillarum*	0.25	5.08	19.76	0.81	1.49	28.62	0.86	0.91	27.10
*A.salmonicida vs Y.ruckeri*	0.24	5.84	19.69	0.82	1.65	28.07	0.69	1.94	27.25
*P. salmonis* vs *V. anguillarum*	0.27	4.58	11.41	0.68	2.07	17.95	0.56	2.83	10.64
*P. salmonis vs Y.ruckeri*	0.25	5.66	11.89	0.68	2.21	20.70	0.58	2.53	11.14
*V. anguillarum vs Y.ruckeri*	0.75	2.41	53.08	0.88	0.90	31.81	0.67	2.07	26.34

**Table 4 vaccines-10-00473-t004:** Sequence and structural alignment of the common secreted antigens. QH, structural homology; RMSD, root mean square deviation.

	Flagellar Hook Assembly Protein FlgD			Flagellar Basal-Body Rod Protein FlgG		
Bacteria Pathogen	QH	RMSD	Identity (%)	QH	RMSD	Identity (%)
*M.viscosa* vs. *A. salmonicida*	0.81	2.95	15.89	0.71	2.35	30.69
*M. viscosa* vs. *P. salmonis*	0.88	1.97	19.18	0.63	2.72	21.06
*M.viscosa* vs. *V. anguillarum*	0.86	2.18	25.17	0.76	1.89	29.31
*M.viscosa* vs. *Y.ruckeri*	0.75	3.19	13.90	0.74	1.80	29.84
*A.salmonicida* vs. *P. salmonis*	0.80	3.03	16.55	0.72	1.63	21.74
*A.salmonicida* vs. *V. anguillarum*	0.80	2.70	18.48	0.81	1.49	28.62
*A.salmonicida* vs. *Y.ruckeri*	0.82	2.16	14.85	0.82	1.65	28.07
*P. salmonis* vs. *V. anguillarum*	0.87	1.64	21.16	0.68	2.07	17.95
*P. salmonis* vs. *Y.ruckeri*	0.73	2.84	15.59	0.68	2.21	20.70
*V. anguillarum* vs. *Y.ruckeri*	0.66	2.59	17.83	0.88	0.90	31.81

**Table 5 vaccines-10-00473-t005:** Common epitopes identified in the five bacterial pathogens and their binding affinities and dissociation constants to Atlantic salmon and Lumpfish MHC II.

Protein	Common Epitopes	N^o^ of Residues	Atlantic Salmon		Lumpfish	
			K_d_ (M) at 10.0 °C	ΔG (kcal mol^−1^)	K_d_ (M) at 10.0 °C	ΔG (kcal mol^−1^)
LPS assembly protein LptD	PYYLNLAPNYD QTLEPRLYYLYVP	11 13	2.3 × 10^−7^ 2.3 × 10^−7^	−8.6 −8.6	2.5 × 10^−7^ 5.8 × 10^−8^	−8.6 −9.4
Outer membrane protein assembly factor BamA	IEGLQRL DEVLRR LNTLGYF LQWMSPLGP	7 7 7 10	1.5 × 10^−6^ 9.4 × 10^−6^ 5.1 × 10^−7^ 7.9 × 10^−7^	−7.5 −6.5 −8.1 −7.9	5.5 × 10^−6^ 9.4 × 10^−6^ 2.2 × 10^−7^ 1.9 × 10^−7^	−6.8 −6.5 −8.6 −8.7
TonB-dependent siderophore receptor	EKIDVRGGAAVQYG	15	5.1 × 10^−9^	−10.3	6.8 × 10^−8^	−9.3
Flagellar hook assembly protein FlgD	LKNQDPPTNP LQASALLG WDGNDQNGN	10 8 9	1.8 × 10^−7^ 1.9 × 10^−7^ 2.5 × 10^−8^	−8.7 −8.7 −9.8	5.4 × 10^−9^ 6.4 × 10^−7^ 1.5 × 10^−7^	−10.7 −8.0 −8.8
Flagellar basal-body rod protein FlgG	LLTQLAQQDP ALQASALVG WDGKL	10 9 5	1.1 × 10^−7^ 1.6 × 10^−7^ 1.4 × 10^−6^	−9.0 −8.8 −7.6	8.0 × 10^−8^ 4.0 × 10^−7^ 4.7 × 10^−6^	−7.9 −8.3 −6.9

## Data Availability

Not applicable.

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
