# Peer review of "Comparative Reverse Vaccinology of *Piscirickettsia salmonis*, *Aeromonas salmonicida*, *Yersinia ruckeri*, *Vibrio anguillarum* and *Moritella viscosa*, Frequent Pathogens of Atlantic Salmon and Lumpfish Aquaculture"

_vaccines, 2022, doi:10.3390/vaccines10030473_

Round 1
Reviewer 1 Report
The manuscript n. 1607616 deals with a list of potential antigens for designing new protocols for polyvalent vaccination in fins-fish focused on salmonid species. I believe in this strategy of polyvalent immunisation; however, the authors should cite other research in the field because the vaccination strategy is relatively new. There are more than 15 years that the researchers are voted in this strategy!. Moreover, the bioinformatic model has been built to optimise the polyvalent vaccination strategy based on fish species' size/age and immune response. The combination of potential antigens and the application of in silico model could be the key to understanding the efficacy of possible vaccination before the in vivo use. The authors should take into count these manuscripts in their work. The manuscript needs a minor English revision.
Title: title should be changed because it is focused on salmonids, and only a few antigens are related to other marines/freshwater fish species. I suggest: "Comparative Reverse Vaccinology of Frequent Pathogens of Salmonids and other Finfish species in Aquaculture". The name of antigens studied can be inserted in the keywords. The title should be short!
Introduction:
Lines 56-65 The authors should read and cite in their work the in silico model of polyvalent vaccination and other polyvalent typology of vaccination in salmonids (salmon and trout) and at least in several seawater species (sea bass, sea bream):
Semin. Immunol. (2004), 16, 335–347
Fish Shellfish Immunol. 2007 Mar;22(3):206-17. doi: 10.1016/j.fsi.2006.04.010.
Vaccine, 2013 Feb 6;31(8):1224-30. doi: 10.1016/j.vaccine.2012.12.041.
J Veterinary Sci Technol 2016, 7:2 http://dx.doi.org/10.4172/2157-7579.1000299
Bioinformatics, Volume 33, Issue 19, 01 October 2017, Pages 3065–3071, https://doi.org/10.1093/bioinformatics/btx341
Microorganisms. 2019 Nov 29;7(12):627. doi: 10.3390/microorganisms7120627.
Microorganisms 2019, 7(11), 569; https://doi.org/10.3390/microorganisms7110569 -
Results/discussion.
Next to the bioinformatic study, to validate the results and the best epitopes found to develop a polyvalent vaccination, it needs at least proof with the mathematic model with "in silico" fish. Thus, I suggest inserting a validation of two or three more promising antigens with the mathematical model that I have already suggested.
Reviewer 2 Report
The Manuscript ID vaccines-1607616 entitled "Comparative Reverse Vaccinology of Piscirickettsia salmonis, Aeromonas salmonicida, Yersinia ruckeri, Vibrio anguillarum and Moritella viscosa, Frequent Pathogens of Marine Finfish Aquaculture", submitted for publication in Vaccines, deals with a very interesting topic in aquaculture such as the prevention of infectious diseases by vaccination.
The manuscript uses a very innovative methodology to identify and select possible common antigens to several pathogenic bacteria species important in marine aquaculture in order to develop a polyvalent vaccine. Although the reviewer is not an expert in this methodology and is unaware of the correlation that could exist between predictive findings using this methodology and in vivo effectiveness, I believe that the results obtained in this study are very interesting and I strongly recommend its publication.
The reviewer only suggests some minor changes:
Line 232: the sentence "hypothetical proteins were not included in the analysis" should be explained (I am not sure what it´s mean).
Line 298: I think the text refers to figure 9 not figure 8.
Lines 359-362: rewrite because there are repetitions.
Line 472-475: why lumpfish is mentioned, it is the first time throughout the text; it wouldn't be atlantic salmon instead.
Author Response
Reviewer #2
The Manuscript ID vaccines-1607616 entitled "Comparative Reverse Vaccinology of Piscirickettsia salmonis, Aeromonas salmonicida, Yersinia ruckeri, Vibrio anguillarum and Moritella viscosa, Frequent Pathogens of Marine Finfish Aquaculture", submitted for publication in Vaccines, deals with a very interesting topic in aquaculture such as the prevention of infectious diseases by vaccination.
The manuscript uses a very innovative methodology to identify and select possible common antigens to several pathogenic bacteria species important in marine aquaculture in order to develop a polyvalent vaccine. Although the reviewer is not an expert in this methodology and is unaware of the correlation that could exist between predictive findings using this methodology and in vivo effectiveness, I believe that the results obtained in this study are very interesting and I strongly recommend its publication.
We appreciate the reviewer’s kind comment
Line 232: the sentence "hypothetical proteins were not included in the analysis" should be explained (I am not sure what it´s mean).
RE: We have provided a description for hypothetical protein.
Line 190: “Hypothetical proteins which are proteins not yet characterized on NCBI”
Line 298: I think the text refers to figure 9 not figure 8.
RE: Thanks! This has been fixed.
Lines 359-362: rewrite because there are repetitions.
RE: Thanks! This has been fixed
Line 472-475: why lumpfish is mentioned, it is the first time throughout the text; it wouldn't be Atlantic salmon instead.
RE: It was an oversight. However, we have expanded the study to cover both lumpfish and Atlantic salmon
Round 2
Reviewer 1 Report
The authors in the rebuttal letter have hardly followed the suggestions from my first revision.
The authors should give a list of the predicting antigens to use in vaccination and tests that have proven their found, or they can change the title with something that describes the content of this manuscript. Moreover, since the authors have included the lumpfish's data to justify the title deals of "marine scìpecies", it is still confusing with the actual content of the results. It is correct to include another seawater fish; however, the lumpfish has still related to the salmon culture. Lumpfish have been used as biological control of sea lice in the salmon seawater culture. This method was developed in Norway and has since gained popularity with salmon producers, particularly in Canada, which has increased the cleaner fish production capacity in the last five years. Thus, I can believe the interest of these Canadian authors in these two species. However, because of this focus, I still suggest changing the title in another with a better sound:
"Comparative reverse vaccinology of some common pathogens for predicting a polyvalent vaccine in salmonid breeding"
Author Response
Dear Reviewer, we followed your comments the best that we could understand and fit them in the context of the article. We are providing Supplementary Table S4, which lists all the antigens that could be used for vaccine development.
The suggested title for the reviewer, still dost not fit our research, in fact, puts us in a very difficult situation since creates a strong misleading.
Lumpfish and salmon are very different species and although they are co-cultivated, they do not have the same breeding. Therefore the title suggested by the reviewer "Comparative reverse vaccinology of some common pathogens for predicting a polyvalent vaccine in salmonid breeding" is not representing our research and adds confusion to the readers as is not precise.
We modified the title to "Comparative Reverse Vaccinology of Piscirickettsia salmonis, Aeromonas salmonicida, Yersinia ruckeri, Vibrio anguillarum and Moritella viscosa, Frequent Pathogens of Atlantic Salmon and Lumpfish Aquaculture."